# Transcriptomic Profiling of Quinoa Reveals Distinct Defense Responses to Exogenous Methyl Jasmonate and Salicylic Acid

**DOI:** 10.3390/plants14111708

**Published:** 2025-06-03

**Authors:** Oscar M. Rollano-Peñaloza, Sara Neyrot, Jose A. Bravo Barrera, Patricia Mollinedo, Allan G. Rasmusson

**Affiliations:** 1Department of Biology, Lund University, Kontaktvägen 13, SE-223 62 Lund, Sweden; 2Dirección de Investigación, Ciencia y Tecnología (DICyT), Universidad Mayor Real y Pontificia de San Francisco Xavier de Chuquisaca, Rosendo Villa 150, Sucre, Bolivia; 3Instituto de Investigaciones Químicas, Universidad Mayor de San Andrés, Campus Universitario Cota Cota c 27, La Paz P.O. Box 12958, Bolivia

**Keywords:** jasmonic acid, next-generation sequencing, plant hormones, quinoa, RNA-seq, salicylic acid, transcriptomics, *Trichoderma*

## Abstract

Plant defense responses are mediated by hormones such as jasmonic acid (JA) and salicylic acid (SA). JA and SA are known to trigger a range of different defense responses in model plants but little is described in crops like quinoa. Here, we present the first molecular description of JA and SA signaling at the transcriptomic level in quinoa. The transcriptomes of quinoa cv. Kurmi seedlings treated with 100 µM methyl JA or 1 mM SA for 4 h were analyzed, using on average 4.1 million paired-end reads per sample. Quinoa plants treated with JA showed 1246 differentially expressed (DE) genes and plants treated with SA showed 590 DE genes. The response to JA included the induction of genes for the biosynthesis of JA (8/8 genes) and lignin (10/11 genes), and displayed a strong association with treatments with *Trichoderma* biocontrol agents. The SA treatment triggered the upregulation of genes for the biosynthesis of monoterpenoids and glucosinolates, both having defense properties. Overall, this suggest that JA and SA promotes the biosynthesis of lignin polymers and chemical defense compounds, respectively. Overall, the DE genes identified can be used as molecular markers in quinoa for tracking plant-hormone pathway involvements in defense responses.

## 1. Introduction

Quinoa (*Chenopodium quinoa* Willd.) is an allotetraploid crop that has gained wide international interest due to its nutritional properties; its seeds and greens are gluten-free with a high protein content (15%), containing all essential aminoacids, vitamins (B_1_, B_2_, B_3_, C and E), minerals (Ca, P, Mg, Fe, Zn, K, Cu), antioxidants, and fatty acids desired for human nutrition [1,2,3]. Quinoa also possesses a high tolerance to drought and salinity [4,5,6] which has contributed to its cultivation in more than 120 countries across five continents [2]. Therefore, quinoa has been extensively investigated regarding its capability to tolerate abiotic stress [7]. In contrast, its response to biotic stress has been far less studied, despite the fact that plant diseases (e.g., downy mildew) [8] and insect plagues (e.g., quinoa moth) are the main factors affecting crop yields in the Andean highlands, the native homeland of quinoa [9]. For example, the downy mildew disease caused by the oomycete *Peronospora variabilis* can lead to crop losses up to a 99% in susceptible cultivars (e.g., cv. Utusaya) unless controlled [10]. However, quinoa display a great genetic diversity (several thousand accessions) [11], where we can observe quinoa cultivars that display tolerance to *P. variabilis* such as quinoa cv. Kurmi [12,13,14]. Quinoa cv. Kurmi has been shown to have a higher tolerance to diseases by delaying pathogen sporulation and by increasing the production of pigments such as betalains [12,15,16].

Plant defense responses, mainly based on observations in *Arabidopsis thaliana*, are regulated by phytohormones such as salicylic acid (SA) and jasmonic acid (JA). Plant defense against biotrophic pathogens usually involves the activation of SA-mediated defense response pathways. On the other hand, defense response against necrotrophic pathogens and insects involves JA to a greater extent [17]. The crosstalk between the signaling pathways of these two hormones creates a highly regulated network to prevent pathogen damage and optimize the defense response. SA usually takes transcriptional control over JA [18], but SA-mediated defense responses can also be regulated and even suppressed by JA [19]. Recent discoveries have shed more light on the impact of JA on the defense response against biotrophic pathogens; in grapevine, it has been shown that resistance to the biotrophic pathogen *Plasmopara viticola* is mediated by JA [20]. The application of JA to plants related to quinoa, such as sugarbeet, have shown to reduce the progression of disease symptoms by multiple post-harvest pathogens [21]. Nevertheless, most of these studies have been analyzed at gene-by-gene level and only a few non-model plants have been studied at the transcriptomic level to understand the function of plant hormones.

Quinoa exposure to JA and SA has been used to overcome drought stress symptoms [22]. The effect of JA application in quinoa has also been studied to evaluate JA impact on saponin biosynthesis through gene-expression changes [23]. However, little is known about the effect of JA and SA for quinoa to overcome biotic stress. Previous investigations of quinoa-microbial interactions have indicated associations to quinoa orthologs of genes that are responsive to biotic stress hormones such as JA [24], but a direct association of quinoa genes to hormones has not been determined.

In order to provide a stable foundation for transcriptomics analyses of quinoa biotic interactions, and avoiding analyses by proxy, we present here the quinoa transcriptomic response to treatments with JA and SA. The results suggest that quinoa seedling exposure to JA in vitro triggers genes for the synthesis of JA and the production of lignin monomers. Meanwhile, the application of exogenous SA suggested the upregulation genes for monoterpenoid biosynthesis and SA degradation.

## 2. Results

### 2.1. Transcriptomic Analysis of Quinoa Seedlings Treated with JA or SA

Quinoa cv. Kurmi was selected for the study due to its higher tolerance against *P. variabilis* as compared to other cultivars tested. Five RNA samples from 2 days-old quinoa seedlings treated with SA or methyl JA (hereon termed JA) for 4 h (h) were collected. The four RNA samples that showed best quantity and quality parameters were selected for RNA-seq. Sequencing was carried out with an output between 3.87 and 4.53 million paired-end reads of 300 bp length per sample (Table 1). Reads were mapped to the chromosome-level assembly of the quinoa genome cultivar QQ74 (Table 1). The proportion of mapped reads to the QQ74 quinoa genome was on average 95.4%.

Reads aligned to a single gene accounted for 13.8%, reads aligned to two genes (ambiguous) were only 0.017%. Reads that were aligned to multiple genes with no clear assignment (alignment_not_unique) were 34%. The majority of reads were defined as “no feature” (48.5%) during counting with HTseq; therefore, we suggest those reads belong to noncoding RNA because we used a CDS reference annotation file. For these reasons, our downstream analysis was performed only with reads assigned to a single gene, especially for the differential gene-expression analysis (Appendix A).

DGE analysis was performed comparing the transcriptome of each treatment (JA and SA) against the control. Genes that have at least one read count were considered (36,049 genes). Quinoa plants treated with JA showed 1246 DE genes (Appendix A), while quinoa plants treated with SA showed 590 DE genes (Appendix A).

### 2.2. KOBAS-i Gene Set Enrichment Analysis of Quinoa Treated with JA and SA

We analyzed JA and SA set of DE genes and we found an overlap of 119 genes that were significantly differentially expressed by both treatments (Figure 1A, Appendix A). There was a higher number of DE genes after JA treatment (1125) as compared to SA treatment (471). JA induced 918 genes, while it repressed 328 genes. SA induced 389 genes, while it repressed 201 genes (Figure 1A).

The bubble-plot gene set enrichment analysis has shown that JA and SA response has the same two pathways with the highest *p* values of enrichment: the biosynthesis of secondary metabolites and metabolic pathways. Both JA and SA samples also displayed significant enrichment for alpha-linolenic acid metabolism; yet, whereas JA induced effects on aromatic amino acid biosynthesis, SA was significantly associated with the metabolism of glutathione, ABC transporters, and porphyrins (Figure 1B,C).

Bar plots on the other hand, as they show the enrich ratio, which measures the relevance for a gene set to an experimental condition, indicate that quinoa treated with JA activates the synthesis and degradation of ketone bodies, alpha-linolenic acid, and linoleic acid metabolism as the top categories (Figure 1D). Quinoa treated with SA revealed that the most enriched genes are involved in the biosynthesis of glucosinolates, monoterpenoid biosynthesis, and ABC transporters (Figure 1E).

### 2.3. KOALA Annotation and KEGG Mapping Reveals That JA Application Induces Genes for Biosynthesis of Jasmonates

The sets of differentially expressed genes were further analyzed for the annotation of genes for metabolic pathways in the KEGG database with BlastKOALA. Thus, KEGG enrichment analysis for JA showed that alpha-linolenic metabolism genes were highly enriched (Figure 1D). Linolenic acid is a precursor of JA and is involved in its biosynthesis. Looking at the KEGG enrichment in more detail, we could observe that JA induced the gene expression of eight quinoa genes (AUR62019248, AUR62019249, AUR62012258, AUR62002817, AUR62000076, AUR62000077, AUR62009902 and AUR62001467) that were annotated as orthologs of lipoxygenase-2 (LOX2, K00454). LOX2 produces 13(S)-hydroxiperoxyoctadeca-9,11-dienoic acid (13S-HPODE), a precursor of JA. In fact, the whole JA biosynthesis KEGG module (M00113) was significantly induced (Figure 2, Table 2, and Appendix A). The automatic annotation by KEGG pathway map viewer can be seen in Appendix A. Thus, the results suggest that JA was synthesized from alpha-linolenic acid in quinoa when the plants were spray-treated with JA, in what may be a positive feedback loop regulation. LOX1 was also significantly induced in quinoa, a gene that was previously thought to be involved in JA biosynthesis through the linoleic acid metabolism (K15718, AUR62005156).

### 2.4. Quinoa Treatment with JA Induces Genes Involved in Lignin Biosynthesis

BlastKOALA annotated almost half (618 genes; 49.6%) of the whole set of DE genes responsive to JA treatment to functional categories. The highest proportion of genes was annotated to the large gene groups associated with carbohydrate metabolism (15%), amino acid metabolism (11.5%), and signaling and cellular processes (10.5%) (Figure 1). The annotation by KOALA has better correlation with the enriched bubble plot generated by Kobas-i than the bar plot enrichment (Figure 1).

In order to pinpoint whole signaling pathways induced by JA treatment in quinoa, a gene enrichment was performed with the KEGG Mapper. KEGG reconstruction showed that five pathway modules were differentially expressed and were the following (Appendix A): JA biosynthesis and beta-oxidation in the acyl-CoA synthesis as part of the lipid metabolism module (M00087); the serine and cysteine biosynthesis in the amino acid metabolism module (M00021); the mevalonate pathway (C5 isoprenoid biosynthesis) in the module of biosynthesis of terpenoids and polyketides (M00095); and the module of flavanone biosynthesis (M00137) that converts phenylalanine into naringenin inside the biosynthesis of the other secondary metabolites module.

The module’s signature inside the KEGG mapper reconstruction analyzes whole biological pathways where the software can annotate all or some DE genes that belong to a certain pathway. The module signature allows us to explore biological pathways with 1 or more components (blocks) that are not annotated (i.e., a gene missing on a DGE list of a pathway during a certain experiment). During our JA treatment of quinoa plants, we found that several genes in the monolignol biosynthesis pathway were differentially expressed (Figure 3, Table 3). Putative quinoa genes for 10 out of 11 enzymes involved in the production of the lignin monomers sinapyl and p-coumaryl, and coniferyl alcohols were significantly induced (Table 3, Appendix A). The automatic annotation by the KEGG pathway map viewer can be seen in Appendix A. Among them, genes for enzymes that produce non-conventional lignin monomers such as guaiacyl, syringyl, 5-OH—guaiacyl, and p-hydroxyphenyl units were also induced. This suggests that lignin synthesis may be generally elevated during treatment with external JA.

### 2.5. JA May Induce the Biosynthesis of Saponins

Saponins are synthesized from oleanolic acid which is the product of B-amyrin monooxygenases (*CqbAMO1* and 2) and B-amyrin synthases (*CqbAS1* and 2). We found two B-amyrin synthases differentially induced; one of them was the gene *CqbAS1* (AUR62025693) that was previously reported to be induced by JA treatment in quinoa plants [23], and here we also report the DE of its ortholog (*CqbAS2*, AUR62001311), which has a 89.3% identity at the genomic level and 96.6% at protein level. Our DGE analysis also showed B-amyrin monooxygenases *CqbAMO1* (AUR62025699) and *CqbAMO2* (AUR62001317), previously described as P450 enzymes (*CqCYP716A78* and *CqCYP716A79*). We also found two genes for the quillaic acid 3-O-glucuronosyltransferase which are involved in the saponin adjuvant biosynthesis pathway. Altogether, the differential induction of these genes indicates that saponins might be synthesized upon JA treatment.

### 2.6. Other Pathways Differentially Expressed by JA Treatment in Quinoa

JA induced the expression of several additional gene groups. Related to redox metabolism and oxidative stress, nine genes of glutathione-S transferase GST (K00799) were DE, where two genes were repressed and seven were induced. We also observed a significant induction of seven ascorbate peroxidases (K00434). While analyzing chalcone synthases (K00660), we observed that two were induced and one was repressed. The analysis of chitinases (K01183) showed two repressed and one induced. Finally, we observed seven acetyl methyl transferases (K22588) that were all induced (Appendix A).

### 2.7. Quinoa Differential Gene Expression in Response to SA Treatment

The treatment of quinoa plants with SA lead to 590 genes differentially expressed. Among them, we observed four genes significantly induced which are annotated in the monoterpenoid biosynthesis pathway by KEGG. Quinoa gene AUR62004316 was annotated as neomenthol dehydrogenase (K15095) and three genes (AUR62003693, AUR62024168 and AUR62003695) were annotated for 8-hydroxygeraniol dehydrogenase (K23232). In addition, other genes were annotated in the metabolism of terpenoids, specifically carotenoids, where we observed one gene induced (AUR62019188; K02291) and another repressed (AUR62031420; K15746), suggesting that carotenoids are synthesized during SA treatment.

Among defense response genes that were differentially expressed by SA, we observed seven peroxidases significantly induced (K00430), thirteen glutathione S-transferases differentially expressed (K00799) where twelve were induced and one was repressed, and five DE chitinases (K01183) with four being induced and one repressed. Regarding metabolic pathways related to defense response, we observed four DE lipoxygenases (K00454), two induced and two repressed. Interestingly, we observed that all five DE chalcone synthases (K00660) were repressed, contrarily to what we observed with JA where two were induced and one repressed (Appendix A). We could observe the significant induction of two putative salicylic acid 5-hydroxylase (*AtSA5H*) genes: AUR62028083 (*CqSA5HA*) and AUR62005987 (*CqSA5HB*) (Appendix A).

Finally, in order to analyze the correlation of genes expressed by hormone (JA and SA) treatment and the biocontrol agent *Trichoderma* spp., we compared the DGE list of quinoa plants treated with JA and SA to the treatment of the same quinoa cultivar (cv. Kurmi) with two *Trichoderma* species: *Trichoderma harzianum* BOL-12 (T.BOL12) and *Trichoderma afroharzianum* T22 (T22) [24]. The response of quinoa to JA shared a statistically significant association with the expression of 28 DE genes with T.BOL12 and 234 DE genes with T22 (Figure 4). The response of quinoa to SA shared a statistically significant association only with *Trichoderma* strain BOL12 (13 DE genes). Nevertheless, the response of quinoa to SA shared the expression of 83 DE genes with T22 (Figure 4 and Appendix A). Overall, data thus indicate a clear association between the quinoa response to JA and the response to *Trichoderma* spp.

## 3. Discussion

Transcriptomic analyses of quinoa seedlings collected 4 h after JA treatment resulted in the identification of 1246 DE genes, and after SA treatment we identified 590 DE genes. However, we have to consider that these results involve whole RNA, where there might be a bias towards noncoding RNA. The majority of reads during counting with HTseq were defined as “no feature” (48.5%); therefore, we suggest “no feature” reads belong to noncoding RNA because we constructed the sequencing library with a Truseq stranded total RNA kit and we used a CDS reference file (GTF file) for mapping and counting. We also identified that a proportion of reads (34%) were tagged as non-unique due to the inability of the software to properly identify the right gene to count. This could be attributed to the nature of the quinoa genome, which is tetraploid; many genes are duplicated or have a similar pseudogene, which makes the read assignation to a single gene difficult [25]. Despite the above, we had enough reads (average = 565.577 reads) to show whole DE metabolic pathways (Table 1).

Among the 1246 DE genes, we identified the upregulation of several quinoa gene orthologs of genes known to be responsive to JA (Appendix A). These include two quinoa orthologs (AUR62018713 and AUR62007294) of the *Arabidopsis thaliana AtMYC2* gene, a well-known JA-responsive transcription factor in *Arabidopsis* [26], which can confirm that response of quinoa to JA is somewhat similar to what is reported in *A. thaliana* [26] (Appendix A) and tomato [27]. The response of our quinoa plants to JA also includes several genes previously reported to be upregulated by JA in quinoa [23]. For example, we observed an upregulation of B-amyrin synthase (*CqbAS*; AUR62025693) and two cytochrome P450 monooxygenases (*CYP450*; AUR62025699 and AUR62001317) that were previously shown to be DE by drenching mature quinoa leaf tissue in JA [23].

During the JA treatment of quinoa plants, we identified several DE genes related to jasmonate biosynthesis (Figure 2). This is consistent with previous observations in *A. thaliana*, where JA creates a positive feedback regulation of JA biosynthesis and it is described as a primary response of JA signaling [28,29]. The generally accepted view is that a positive feedback keeps JA levels high enough to keep the plant’s response activated against long-term stress [28,30,31].

One of the main defense responses in plants is lignification [32]. However, how the lignin biosynthesis pathway is activated is not completely clear. We observed that many genes in the lignin biosynthesis pathway have been differentially expressed by JA treatment in our analysis, which might indicate its activation. The induction of lignification by methyl jasmonate has been reported in *Arabidopsis* cell cultures [33], tea plants [34], *A. thaliana*, and poplar plants by mutant analysis and gene expression [35], as well as monocots such as *Brachypodium distachyon* callus [36] and gymnosperms such as conifers from the Pinaceae family [37].

The quinoa lignin biosynthesis seems to share the core molecular components for biosynthesis of conventional lignin monomers [38,39]. Given that 15 peroxidases (K00430) that synthesize lignin alcohol monomers such as guaiacyl, syringyl and p-hydroxyphenyl were expressed (Figure 3), we could assume that this set of enzymes catalyzes the oxidation of the basic lignin alcohol monomers to produce the lignin subunits which ultimately form the lignin polymer. Peroxidases have been observed to be DE during quinoa infections with *P. variabilis*, gathering evidence on their role in defense responses [40].

JA may induce the synthesis of saponins in quinoa plants, because the spray treatment of quinoa with JA induced several genes present in the biosynthetic pathway of triterpene saponins from 2,3-oxidosqualene, as previously reported [23]. Among them, we could observe B-amyrin monooxygenases (*CqbAMO1* and *2*, Appendix A) which convert B-amyrin into oleanolic acid that will ultimately be converted to triterpene saponins. Besides the aforementioned enzymes, other putative B-amyrin monooxygenases (AUR62023556 and its ortholog AUR62004592) were also DE, and these enzymes might be involved in the conversion of B-amyrin into oleanolic acid. JA has been shown to enhance saponin biosynthesis in medicinal herbs such as *Panax notoginseng* [41] and *Dioscorea zingiberensis* [42].

Finally, JA upregulated the expression of quinoa genes involved in the antioxidant response such as L-ascorbate peroxidase, annexins, and catalase. Annexins have been described to be involved to the abiotic and biotic stress response in plants [43]. Annexins can reduce the oxidative damage in the plant and allow for tissue recovery [44].

Quinoa differential gene expression to SA treatment showed that genes known to be responsive to SA treatment in other species, such as the WRKY70 transcription factor family [45,46], were DE in quinoa (Appendix A). However, known SA-induced genes such as the Pathogenesis-related protein (*AtPR1*; AT2G14610) and Non-expressor of Pathogen Resistance gene (*AtNPR1*; AT1G64280) were not only not expressed but clear orthologs could not be identified in quinoa.

SA is known to stimulate plant defenses through the biosynthesis of chemical compounds such as monoterpenoids, both by internal biosynthesis [47] and through exogenous application [48,49,50]. Therefore, the expression of genes annotated for the biosynthesis of monoterpenoids in quinoa upon SA treatment was expected (Figure 1E). The gene set enrichment analysis showed that SA application caused the upregulation of the glucosinolate biosynthesis pathway (Figure 1E). Although only 2 quinoa genes encoding for UDP glycosyltransferases (AUR62039410 and AUR62026575) were annotated for glucosinolate biosynthesis, it is expected that the whole pathway might be upregulated upon exogenous SA application, as described by Halkier and Du [51]. This indicates that the application of exogenous SA might contribute to improve quinoa defenses through the enhanced expression of monoterpenoid and glucosinolate biosynthesis.

The upregulation of two genes AUR62028083 (*CqS5HA*) and AUR62005987 (*CqS5HB*) that putatively encode salicylic acid 5-hydroxylases (*AtS5H*) suggests that salicylic acid is actively degraded in quinoa upon exogenous application. *AtS5H* is an enzyme that degrades SA into 2,5-dihydroxybenzoic acid (2,5-DHBA or gentisic acid) [52]. This enzyme fine-tunes SA homeostasis, and the downregulation by the knockdown or knockout of this gene could enhance the defense response of quinoa, as reported previously for *Arabidopsis* [52], peanut [53], and tomato [54]. In fact, the disruption of SA 5-hydroxylases in rice enhances its resistance against several pathogens, and the overexpression of SA-degrading enzymes increases the susceptibility of rice plants to pathogens [55]. SA levels can fluctuate very much between plants and even between cultivars, as described in rice [56]. Therefore, we suggest that the high overexpression of *CqSA5HA-B* of quinoa could reduce the effect of exogenous SA. Further, an overexpression of *CqSA5HA-B* of quinoa might explain the lack of hypersensitive response in this cultivar and its susceptibility to downy mildew disease [12].

The response of quinoa to JA overlaps statistically significant with the response to *Trichoderma* spp. (Figure 4), indicating that *Trichoderma* spp. induces a JA response in quinoa. Previous transcriptomic studies on the response of quinoa to *Trichoderma* spp. were not able to associate the response to JA with Gene Ontology analysis [24]. This might have been because not enough genes were automatically annotated as responsive to JA in non-model species such as quinoa. The association of plant responses to *Trichoderma* to JA was first proposed during experiments with cucumber (*Cucumis sativus*) and JA biosynthesis inhibitors [57]. There, it was also shown that some genes in the JA pathway such as *CsLOX1* and *CsPAL1* were induced by *Trichoderma*. Later, the induction of some JA-related genes by *Trichoderma* was demonstrated in *A. thaliana* [58] and was finally confirmed with transcriptomic studies [59].

The response of quinoa to SA was associated with one *Trichoderma* species but not with the other (Figure 4). The response of plants to *Trichoderma* strains might or might not be associated with SA as has been reported before, showing the response of *Trichoderma atroviride* associated with both JA and SA hormone pathways in *Arabidopsis* [58].

In conclusion, the research presented here has identified whole pathways differentially expressed in quinoa by JA treatment, which opens a new way to identify biological significance out of transcriptomic data. In addition, we can observe that the quinoa transcriptomic data generated in this study can contribute to the association of quinoa defense responses to SA and JA hormone-mediated pathways.

## 4. Materials and Methods

### 4.1. Biological Materials

Seeds of quinoa (*Chenopodium quinoa* Willd.) cultivar Kurmi were kindly supplied by PROINPA (Quipaquipani, La Paz, Bolivia). The study was performed under pertinent institutional, national, and international guidelines and legislation.

### 4.2. Hormone Treatments

Disinfection, germination, and in vitro cultivation of *C. quinoa* seeds were performed as previously described [15]. Five days-old quinoa seedlings (shoot and root) were sprayed to dripping point with a water solution. We applied 100 µM methyl jasmonate (hereon termed JA) (Sigma-Aldrich, St. Louis, MO, USA) (stock solution dissolved in absolute ethanol) and 1 mM of SA (Sigma-Aldrich, St. Louis, MO, USA) (pH = 7.0) dissolved in sterile MiliQ water. Mock treatments used absolute ethanol for JA treatment and MiliQ water for SA.

### 4.3. Sample Collection and RNA Extraction

For RNA extraction, quinoa seedlings were sampled 4 h after hormone treatment. Each treatment enclosed four plate replicates containing five seedlings. To reduce inter-plate variability, five seedlings were pooled from each of the four plates into pre-weighed aluminum foil envelopes. Aluminum envelopes were weighed and immediately shock-frozen in liquid nitrogen. Frozen samples were either processed immediately or stored at −80 °C until RNA extraction. Total RNA was extracted using the RNeasy plant mini kit (Qiagen, Valencia, CA, USA), with the following modification: Frozen tissue samples were thoroughly ground in a precooled mortar with liquid nitrogen. Then, 450 µL of buffer RLT (Qiagen, Valencia, CA, USA) supplemented with B-mercaptoethanol (1%) was added. Grinding continued until samples thawed and were transferred to a 1.5 mL microcentrifuge tube. The rest of the procedure was followed according to Qiagen instructions. Total RNA quantity was determined with a NanoDrop spectrophotometer. DNase treatment was performed with the DNA-*free* kit (Ambion, Carlsbad, CA, USA), following the instructions of the manufacturer. The integrity and quality of the RNA was determined by agarose gel (2%) as previously described [24].

### 4.4. RNA-Seq Library Construction and Sequencing

Total RNA treated with DNase was sent to the LU DNA sequencing facility (Biology Department, Lund University, Sweden) for RNA quality verification, strand-specific cDNA synthesis, library construction (Truseq Stranded Total RNA) and sequencing using a MiSeq sequencer (Illumina Inc., San Diego, CA, USA) in paired-end mode with a read length of 300 bp. Raw sequences have been deposited at the National Center of Biotechnology Information (NCBI) under project accession number: PRJNA SUB11194058.

### 4.5. Transcriptomic Analysis

RNA-seq reads were checked for quality by FastQC (v.0.9.0) and mapped on the quinoa genome “QQ74” [5] by Tophat2 (v.2.2.9). Transcript abundances were assessed with HTSeq (v.0.9.1) with “intersection-nonempty” mode. Gene-expression levels were measured as counts per million (CPM) [60]. Library size normalization was performed using the trimmed mean of *M*-values (TMM) within the R package edgeR (v.3.14.0) [61,62]. CPM were TMM-normalized in order to compensate for library size differences. Differential gene-expression analysis comparing mock-treated samples with samples treated with JA or SA was performed using edgeR with TMM-normalized libraries [60] with a false discovery rate (FDR) of 5% (q < 0.05) [63].

### 4.6. Functional Annotation of Differentially Expressed Genes

Gene set enrichment for differentially expressed genes was performed with the KEGG Orthology-Based Annotation System (KOBAS-i) [64]. KOBAS-I presents two types of results visualization: Bubble plots and bar plots. Bubble plots show the *p*-value of enriched pathways compared to pathways in a genome as a background. Each node size corresponds to six levels of enriched *p*-values from small to large: 0.05, 0.01, 0.001, 0.0001, 1 × 10^−10^ and 0.1 × 10^−10^. Bar plots show enriched pathways in each row, and the length of the bar represents the enrich ratio, which is given by the ratio of the number of input genes to the background gene number. Here, the genes present in the whole quinoa genome are considered the background genes. Later, the whole set of DE amino acid fasta sequences was annotated with BlastKOALA (KEGG Orthology and Links Annotation) against the KEGG (Kyoto Encyclopedia of Genes and Genomes) database to assign KO (KEGG Orthology) number and identify pathways and modules differentially expressed [65]. Pathway visualization was constructed with MIRO web app. Venn diagrams were constructed with Venny [66]. Quinoa DGE correlation of hormone treatment and *Trichoderma* treatment was carried out from data previously published [24].

## 5. Conclusions

The quinoa cv. Kurmi transcriptomic response to JA and SA shares several genes with responses in model plants such as *Arabidopsis*. However, this investigation identifies the particular quinoa homologues that, among the extra genomic setup caused by allotetraploidy, functionally correspond to the previously described *Arabidopsis* genes. As an example, the knowledge of the JA responsive gene set allowed for the identification of strong JA response involvement in quinoa response to biocontrol fungus of the genus *Trichoderma*.

The quinoa response to JA suggests that lignin is biosynthesized perhaps as a structural defense against pathogen penetration. Also, JA itself appears to be biosynthesized in response to exogenous JA application in a feedback loop that would keep defenses active. The gene response to the exogenous application of SA to quinoa plants suggests that potential defense compounds such as glucosinolates and monoterpenoids are being synthesized. The transcriptomic analysis also suggest that SA might be degraded by SA 5-hydroxylases to regulate SA homeostasis, because these enzymes have the highest gene-expression induction.

Overall, the results presented here provides a ground for determining JA and SA involvement in various growth and environmental interaction processes in quinoa with a focus on defense responses.

## Figures and Tables

**Figure 1 plants-14-01708-f001:**
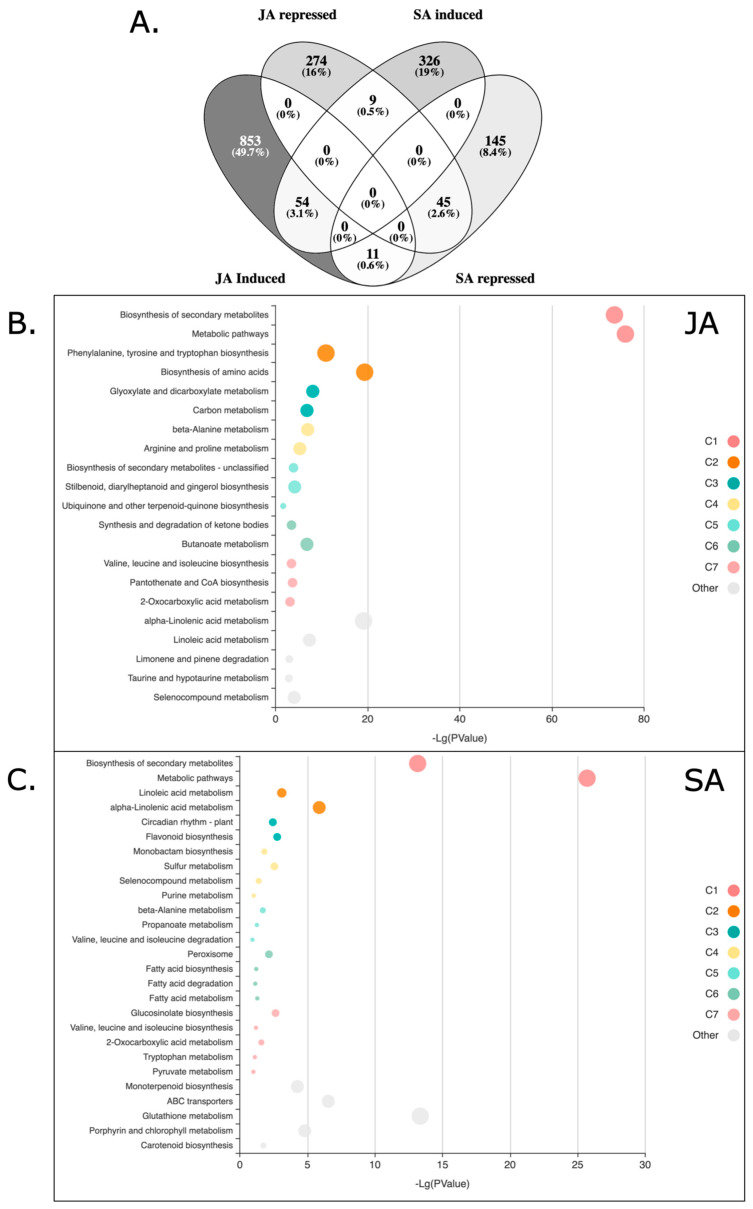
Transcript categories affected by JA and SA. (**A**) Venn diagram of quinoa genes differentially expressed in response to JA and SA. Gene set enrichment analysis bubble plot of JA (**B**) and SA (**C**) signaling. Bubble size is determined by different significance levels. Gene set enrichment analysis bar plot of JA (**D**) and SA (**E**) signaling (continued on next page). Colors in bubble and bar plots reflect clusters (C1–C7) of related metabolic pathways.

**Figure 2 plants-14-01708-f002:**
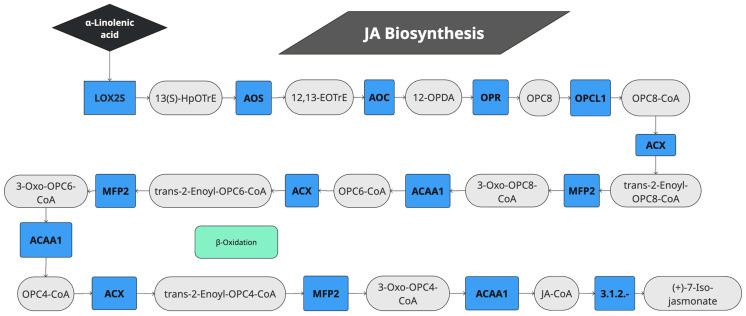
Pathway of jasmonic acid biosynthesis genes activated by JA in quinoa. LOX2S: lipoxygenase; AOS: hydroperoxide dehydratase; AOC: allene oxide cyclase; OPR: 12-oxophytodienoic acid reductase; OPCL1: OPC-8:0 CoA ligase 1; ACOX1: acyl-CoA oxidase; ACX: acyl-CoA oxidase; MFP2: Enoyl-CoA hydratase/3-hydroxyacyl-CoA dehydrogenase; ACAA1: Acetyl-CoA acyltransferase 1; HpOTrE: hydroperoxyoctadeca-9,11,15-trienoic acid; EOTrE: epoxyoctadeca-9,11,15-trienoic acid; OPDA: 12-oxo-phytodienoic acid; OPC: (9S,13S,15Z)-12-Oxo-10,11-dihydrophyto-15-enoate; CoA: Co-enzyme A.

**Figure 3 plants-14-01708-f003:**
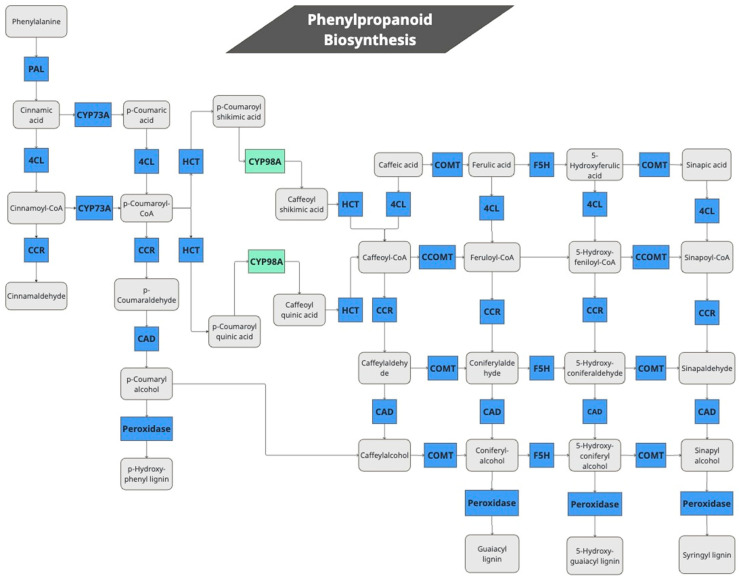
Pathway of lignin biosynthesis genes activated by JA in quinoa. PAL: Phenylalanine ammonia-lyase, CYP73A: Trans-cinnamate 4-monooxygenase, 4CL: 4-coumarat-CoA ligase, CYP98A: 5-O-(4-coumaroyl)-D-quinate 3′-monooxygenase, HCT: Shikimate O-hydroxycinnamoyltransferase, CCOMT: Caffeoyl-CoA O-methyltransferase, COMT: Caffeic acid 3-O-methyltransferase, F5H: Ferulate-5-hydroxylase, CCR: Cinnamoyl-CoA reductase, CAD: Cinnamyl-alcohol dehydrogenase, Peroxidase: peroxidases that produce 4 types of lignin subunits.

**Figure 4 plants-14-01708-f004:**
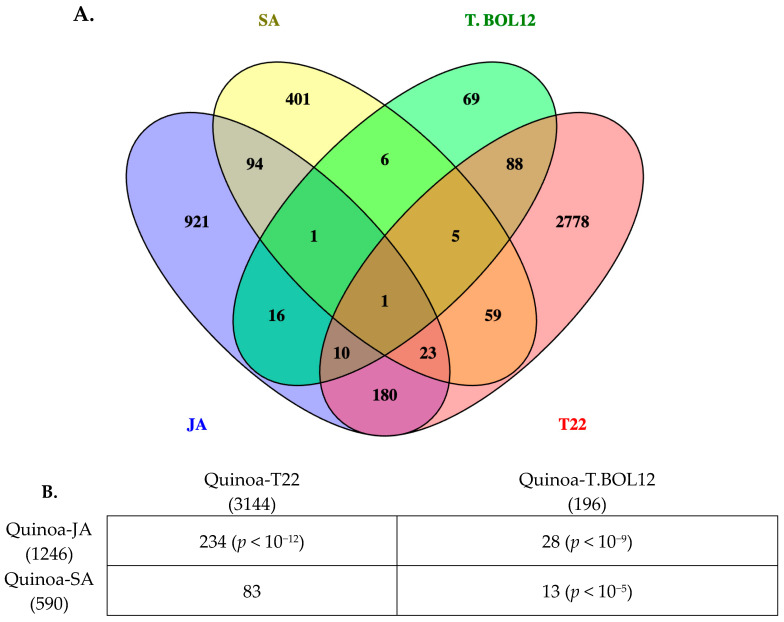
Comparison of quinoa genes that were differentially expressed when treated with JA and SA and when treated with two *Trichoderma* species. (**A**) Quinoa DE genes analyzed with Venn diagrams upon JA and SA treatment compared to *Trichoderma* spp. treatment. (**B**) Number of DE genes shared between hormone (JA and SA) and *Trichoderma* treatment. The total number of genes in each treatment is shown in parentheses. The number of gene overlaps is shown with chi-square *p*-values in parentheses (when less than 0.05).

**Table 1 plants-14-01708-t001:** RNA-seq summary of quinoa sequence reads mapped during SA and JA treatment.

Sample	Treatment	Total Reads ^1^	Mapped Reads	%	Unique Reads ^2^	%	Non-Unique Reads ^3^	%	Noncoding RNA Reads ^4^	%	Ambiguous
1	Control 1	4,030,580	3,983,129	96.1	497,482	12.5	1,328,848	33.4	2,033,141	51.0	622
2	Control 2	4,171,906	4,171,831	94.3	478,990	11.5	1,391,787	33.4	2,142,277	51.4	608
3	Control 3	4,534,183	4,486,832	96.1	561,575	12.5	1,527,014	34.0	2,256,616	50.3	691
4	Control 4	4,297,813	4,240,505	95.7	595,001	14.0	1,349,547	31.8	2,149,134	50.7	750
5	SA 1	4,221,902	4,176,269	96.0	468,141	11.2	1,452,097	34.8	2,119,621	50.8	610
6	SA 2	4,100,965	4,059,417	96.4	477,640	11.8	1,633,839	40.2	1,828,580	45.0	612
7	SA 3	3,984,027	3,935,189	95.1	564,365	14.3	1,307,064	33.2	1,906,205	48.4	785
8	SA 4	4,111,569	4,070,911	95.7	614,441	15.1	1,418,030	34.8	1,891,153	46.5	846
9	JA 1	4,032,015	3,975,797	95.1	656,029	16.5	1,263,491	31.8	1,897,901	47.7	745
10	JA 2	4,376,388	4,327,057	94.9	596,965	13.8	1,495,474	34.6	2,045,298	47.3	664
11	JA 3	3,872,463	3,826,879	95.2	590,513	15.4	1,265,202	33.1	1,818,469	47.5	598
12	JA 4	4,171,328	4,128,924	94.5	685,783	16.6	1,367,084	33.1	1,878,286	45.5	772

^1^ Total reads that passed the quality control. ^2^ Unique reads mapped with Tophat2 and counted with HTseq to the QQ74 coastal quinoa genome. ^3^ Reads mapped to multiple positions with no clear alignment. ^4^ Reads mapped to the genome with no CDS feature annotated.

**Table 2 plants-14-01708-t002:** List of quinoa genes differentially induced upon JA treatment and involved in the α-linolenic acid and jasmonic acid metabolism.

	KEGG Code	Number of Quinoa Genes Induced/Genome Annotated	Enzyme Description	Enzyme Code Figure 2	Enzyme Code
1	K00454	8/14	Lipoxygenase	LOX2S	1.13.11.12
2	K01723	2/7	Hydroperoxide dehydratase	AOS	4.2.1.92
3	K10525	3/6	Allene oxide cyclase	AOC	5.3.99.6
4	K05894	4/10	12-oxophytodienoic acid reductase	OPR	1.3.1.42
5	K10526	2/2	OPC-8:0 CoA ligase 1	OPCL1	6.2.1.
6	K00232	2/7	Acyl-CoA oxidase	ACOX1	1.3.3.6
7	K10527	2/6	Enoyl-CoA hydratase/3-hydroxyacyl-CoA dehydrogenase	MFP2	4.2.1.17
8	K07513	1/4	Acetyl-CoA acyltransferase 1	ACAA1	2.3.1.16

**Table 3 plants-14-01708-t003:** List of quinoa genes differentially induced upon JA treatment involved in lignin biosynthesis.

	KEGG Code	Number of Genes Responsive/Total	Enzyme Description	Enzyme Code Figure 2	Enzyme Code
1	K10775	2/2	Phenylalanine ammonia-lyase	PAL	4.3.1.24
2	K00487	2/4	Trans-cinnamate 4-monooxygenase	CYP73A	1.14.14.91
3	K01904	2/11	4-coumarate-CoA ^1^ ligase	4CL	6.2.1.12
4	K09754	1/6 ^2^	5-O-(4-coumaroyl)-D-quinate 3′-monooxygenase	CYP98A	1.14.14.96
5	K13065	1/22	Shikimate O-hydroxycinnamoyltransferase	HCT	2.3.1.133
6	K00588	2/15	Caffeoyl-CoA O-methyltransferase	CCOMT	2.1.1.104
7	K13066	6/18	Caffeic acid 3-O-methyltransferase	COMT	2.1.1.68
8	K09755	1/1	Ferulate-5-hydroxylase	F5H	1.14.13.-
9	K09753	1/2	Cinnamoyl-CoA ^1^ reductase	CCR	1.2.1.44
10	K00083	1/15	Cinnamyl-alcohol dehydrogenase	CAD	1.1.1.195
11	K00430	15/199	Peroxidases that produce 4 types of lignin subunits.	Peroxidase	1.11.1.7

^1^ CoA: Co-enzyme A. ^2^ Genes showed an at least 1.7-fold increased transcriptomic signal after JA application but not statistically significant.

## Data Availability

Data are contained within the article.

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
