# Peer review of "Transcriptomic Profiling of Quinoa Reveals Distinct Defense Responses to Exogenous Methyl Jasmonate and Salicylic Acid"

_plants, 2025, doi:10.3390/plants14111708_

Round 1

Reviewer 1 Report

Comments and Suggestions for Authors

Plant growth regulators are used in plants for various purposes. Therefore, plant development can be negatively affected by the deficiency or excess of these substances. In this study conducted on the quinoa plant, the effect of BGDM substances was tried to be determined and it is an original article. The references used are sufficient. Modern analysis techniques have been examined and written in a good language. The article outputs contribute to science. However, I have some suggestions.

Latin is missing in the article title

Some numerical data from the article results can be added to the summary.

The summary can be improved a little more.

An introductory paragraph can be detailed about the usage areas, production values, and benefits of quinoa in terms of human health. Because the introduction section is very poorly written.

More data can be added to the introduction section about the lack of function of plant hormones and transcriptomic reactions.

I recommend using 2nd person singular sentences instead of first person plural in line 63.

In line 364, it is written that the study was conducted with the Kurmi variety? Why was this type preferred?

There are many abbreviations in the study, write the full name of the abbreviations when they are first used, or prepare an abbreviation table and give it as an additional file. (For example, instead of h in line 379, hours, etc.)

The concentration of agaros gel should be written in line 392.

The data analysis title is missing in the study.

The results and discussion section of the study is generally well prepared, can more references be added? Just a suggestion.

Also, the figures and tables are well-arranged and visually sufficient.

Only figure 1 is not readable at all. The figure can be broken into pieces and the font can be enlarged.

Similarly, the writings in figure 3 can be clarified.

Can correlation analysis be done in Table 1?

There is a supplementary files title in the article. However, supplementary files have not been uploaded to the system, so this section must definitely be added in the revision. It is difficult to make suggestions and criticisms about this place.

General comment; There is a need for revision in the above-mentioned issues in the study, and the conclusion section is poorly written. It should be updated and rewritten with the parameters that stand out from the study results. Apart from this, I wish the authors success in the revision process.

Author Response

Dear Editor

We are sincerely grateful for the constructive and positive treatment of our manuscript by you and the reviewers. Below are point-by-point responses to all the suggestions, with indications of changes made to the text. We have additionally made some minor text edits to help the understanding. We have also removed the duplicated reference. 

We think that the manuscript has improved substantially and hope that it can now be accepted for publication. 

Best Wishes

Allan G. Rasmusson

Response to Reviewer 1

Plant growth regulators are used in plants for various purposes. Therefore, plant development can be negatively affected by the deficiency or excess of these substances. In this study conducted on the quinoa plant, the effect of BGDM substances was tried to be determined and it is an original article. The references used are sufficient. Modern analysis techniques have been examined and written in a good language. The article outputs contribute to science. However, I have some suggestions.

R: We were very happy to read the positive statement about our work. Thank you for taking your time to review it.

1.    Latin is missing in the article title

R. It has been added, thank you

2.    Some numerical data from the article results can be added to the summary.

R. We added some more numerical data to aid the understanding of the work, Thank you.

3.    The summary can be improved a little more.

R. Thank you, we updated the summary with some more accurate information and also condensed it. We also added a few words on JA - Trichoderma association in the abstract to improve visibility for biocontrol researchers.

4.    An introductory paragraph can be detailed about the usage areas, production values, and benefits of quinoa in terms of human health. Because the introduction section is very poorly written.

R. Thanks for the comment, we agree and because of that we have added a paragraph to incorporate this important information.

5.    More data can be added to the introduction section about the lack of function of plant hormones and transcriptomic reactions.

R. We agree that there is a lack of knowledge on the function of plant hormones involved in biotic interactions and their effects on transcriptome. We have now complemented the introduction with a sentence about the lack of transcriptomic studies on non-model plants regarding plant hormone function.

6.    I recommend using 2nd person singular sentences instead of first person plural in line 63.

R. It has been changed, thank you

7.    In line 364, it is written that the study was conducted with the Kurmi variety? Why was this type preferred?

R: Kurmi was chosen because it was previously shown that is more tolerant then other Andean varieties against the downy mildew disease.

8.    There are many abbreviations in the study, write the full name of the abbreviations when they are first used, or prepare an abbreviation table and give it as an additional file. (For example, instead of h in line 379, hours, etc.)

R. Yes, thank you, we have reviewed carefully to add an explanation at first mention and added them at the abbreviation section. 

9.    The concentration of agaros gel should be written in line 392.

R. It has been added, thank you

10. The data analysis title is missing in the study.

R. Thank you, however, the data analysis done in this study can be found in the “Transcriptomic analysis” section where we described how the data was analyzed.

11. The results and discussion section of the study is generally well prepared, can more references be added? Just a suggestion.

R. Thanks for your suggestion, we have increased some references, specially in the discussion section which we think we have enriched the document.

12. Also, the figures and tables are well-arranged and visually sufficient.

R. Thank you very much!

13. Only figure 1 is not readable at all. The figure can be broken into pieces and the font can be enlarged.

R. Thank you, we have increased the size of the images and re-arranged it so it can be suitable for reading.

14. Similarly, the writings in figure 3 can be clarified.

R. Thanks for the observation, we have increase the font and readability as much as the layout of the journal is allowing us.

15. Can correlation analysis be done in Table 1?

R. Thanks for the suggestion, we have performed a basic correlation and we can see correlation between total and mapped reads (R2= 0,9935). The correlation of the total reads against the other type of reads is not so clear: Unique reads R2= 0,0005, Non-unique reads R2= 0,3456 and Non-Coding RNA reads R2= 0,6072.  

16. There is a supplementary files title in the article. However, supplementary files have not been uploaded to the system, so this section must definitely be added in the revision. It is difficult to make suggestions and criticisms about this place.

R. We apologize for the inconvenient, supplementary files were uploaded in a compressed file but we are uploading them again with the revised version.

17. General comment; There is a need for revision in the above-mentioned issues in the study, and the conclusion section is poorly written. It should be updated and rewritten with the parameters that stand out from the study results. Apart from this, I wish the authors success in the revision process.

R. Thank you very much, conclusions have been rewritten.

Reviewer 2 Report

Comments and Suggestions for Authors

The study presents the first transcriptomic analysis of quinoa responses to JA and SA, identifying key differentially expressed genes involved in defense pathways, including lignin and saponin biosynthesis and monoterpenoid/glucosinolate production, while highlighting potential crosstalk with Trichoderma-induced responses. However, some sections could benefit from additional depth, clarity, or reorganization to enhance readability and impact. Below are specific suggestions for improvement.

1 The title is too general. Please revise it.

2 In the abstract section, the words “results” and so on are unnecessary. Additionally, please simplify the abstract content.

3 Define "Kurmi" cultivar’s agronomic traits in the Introduction to contextualize findings.

4 Clarify whether "methyl-jasmonate" (MeJA) or free JA was used, as these forms differ in bioavailability.

5 The study uses only 4-hour post-treatment samples. Including additional timepoints would clarify whether observed DEGs represent immediate or sustained responses.

6 Ethanol was used as a mock for JA treatments, but its potential confounding effects on gene expression should be addressed. A solvent-only control (water + ethanol) could help isolate JA-specific effects.

7 While KEGG enrichment highlights JA/SA pathways, manual curation of quinoa-specific pathways (saponin biosynthesis) is needed.

8 The upregulation of CqS5HA/B suggests SA catabolism, but enzyme activity assays are needed to confirm this.

9 The experiments for validating the transcriptome results are extremely necessary in the study.

10 The figures are blurry.

Author Response

The study presents the first transcriptomic analysis of quinoa responses to JA and SA, identifying key differentially expressed genes involved in defense pathways, including lignin and saponin biosynthesis and monoterpenoid/glucosinolate production, while highlighting potential crosstalk with Trichoderma-induced responses. However, some sections could benefit from additional depth, clarity, or reorganization to enhance readability and impact. Below are specific suggestions for improvement.

R: We are grateful for the positive evaluation of the manuscript, and hope our improvements are satisfy the expectation

1.    The title is too general. Please revise it.

R. We made the title more specific.

2.    In the abstract section, the words “results” and so on are unnecessary. Additionally, please simplify the abstract content.

R. Thank you, we have removed such words and we simplified the content. 

3.    Define "Kurmi" cultivar’s agronomic traits in the Introduction to contextualize findings.

R. Thank you, a paragraph has been added in the intro and a sentence in the results, thank you.

4.    Clarify whether "methyl-jasmonate" (MeJA) or free JA was used, as these forms differ in bioavailability.

R. Thank you, you made us realize that we didn’t specify in the abstract nor the Results, it has now been corrected.

5.    The study uses only 4-hour post-treatment samples. Including additional timepoints would clarify whether observed DEGs represent immediate or sustained responses.

R. The reviewer is right in that more timepoints would allow more knowledge about the JA and SA responses in quinoa, However, we mainly aimed to identify the response pattern for allowing other researchers to understand better quinoa responses to various biotic plant effectors and signalling compounds.

6.    Ethanol was used as a mock for JA treatments, but its potential confounding effects on gene expression should be addressed. A solvent-only control (water + ethanol) could help isolate JA-specific effects.

R. The amount used for JA treatment was a Me-JA solution composed of 2,24 mg of Me-JA in 100ml of absolute ethanol. Therefore, we decided to use absolute ethanol as the control given the amount of water to be added was minimal compared to the amount of ethanol in the JA treatment.

7.    While KEGG enrichment highlights JA/SA pathways, manual curation of quinoa-specific pathways (saponin biosynthesis) is needed.

R. Thanks for the suggestion, we have re-analyzed DE genes after JA treatment and we have observed that the Saponin adjuvant (QS-7) biosynthesis pathway was partially induced (3/12 genes). In addition to the B-amyrin monooxygenases (CqbAMO1 and 2) described in the text, we observed the induction of two B-amyrin synthase and two quillaic acid 3-O-glucuronosyltransferase. We added a sentence regarding these.

8.    The upregulation of CqS5HA/B suggests SA catabolism, but enzyme activity assays are needed to confirm this.

R. Thank you very much, we agree with this point, that is why we have made only a suggestion and not a hard conclusion, which we expect to generate in the upcoming paper that will be more specific on the effect of SA on quinoa.

9.    The experiments for validating the transcriptome results are extremely necessary in the study.

R. Thank you very much, we agree with this precise point, we wanted to include them in this paper, but we have decided to do it in a new chapter for which we are actually looking for collaborators that can help us validate for example the biosynthesis of JA and the induction of lignification through chemical analysis. In addition, during our previous transcriptomic studies we have observed that RNA-seq was highly correlated to our qPCR analysis and we were advised that it was not necessary this time.

10. The figures are blurry.

R. Thank you for noticing, Figures were updated, specially Figure 1, which had the layout change to increase the Font and improve readability.

Round 2

Reviewer 1 Report

Comments and Suggestions for Authors

Ms is ok

Reviewer 2 Report

Comments and Suggestions for Authors

Accept